# Analyzing the collaborative development needs of grassroots centers for disease control and prevention using the Kano model: A case study of China's Chengdu–Chongqing Economic Circle

Yang Tong[1]*, Huang Qianzhen[1]*, Tan Bo[2], Hu Bin[1], Zhang Min [1]*

**1** Chongqing Shapingba District Center for Disease Control and Prevention, Chongqing, People's Republic of China, **2** Chongqing Center for Disease Control and Prevention, Chongqing, People's Republic of China

\* 75382237@qq.com (ZM); 850084969@qq.com (YT); 26914344@qq.com (HQ)

## Abstract

### Background

Advancing the development of centers for disease control and prevention (CDCs) has become a priority within global public health governance. However, public health governance capacity varies significantly among CDCs across different countries and regions, grassroots CDCs face particular disadvantages. Establishing stable, efficient collaborative development mechanisms among CDCs across diverse regions to maximize overall effectiveness and ensure sustainable development represents a critical public health science issue.

### Objective

This study aims to provide scientific references and a theoretical foundation for the coordinated development of grassroots CDCs within the Chengdu–Chongqing Economic Circle (CCEC) and the construction of public health systems.

### Methods

A questionnaire for collaborative development needs indicators in grassroots CDCs, comprising 4 primary needs and 13 secondary needs, was developed through literature review, the Delphi expert consultation method, and the Kano model. Analysis focused on questionnaires collected from eight grassroots CDCs within the CCEC. The importance of needs was ranked using the better–worse coefficient and satisfaction sensitivity analysis.

### Results

Analysis of the 110 valid questionnaires showed that for the must-be attribute, satisfaction sensitivity ranked as follows: performance compensation (0.883)> talent

**Data availability statement:** All relevant data are within the manuscript and its Supporting Information.

**Funding:** This study was financially supported by the Chongqing Science and Health Joint Medical Research Project (2024MSXM172), and Shapingba District Science and Health Joint Medical Research Project (2024SQKWLHMS032), and Shapingba District Science and Health Joint Medical Research Project (2025SQKWLHMS072)."The funders had no role in study design, data collection and analysis, decision to publish, or preparation of the manuscript.

**Competing interests:** The authors have declared that no competing interests exist.

exchange and scientific research and innovation cooperation (0.824)> public health emergency rescue mechanism (emergency material reserve and cross-regional material mobilization; 0.817)> cross-regional case monitoring, investigation, and tracking (0.775). Regarding the one-dimensional attribute, the satisfaction sensitivity ranking was joint risk assessment and emergency command (0.937)> business archive co-construction and sharing mechanism (emergency response plan, and technical scheme) (0.909)> regional co-construction and sharing between the university and the local area (0.832). For the attractive attribute, the satisfaction sensitivity ranking was regional monitoring and early-warning information management system (0.922)> community chronic disease prevention and service (0.804)> coordinated transfer and diversion diagnosis and treatment of patient with infectious diseases within the region (0.734). However, the collaborative release and interaction mechanism of social integrated media information, public health collaborative governance entities, and the construction of a cross-regional expert database constitute indifferent attributes.

## Conclusions

This study provides preliminary scientific evidence for the precise allocation of public health resources and the establishment of localized collaborative development mechanisms. Simultaneously, the research methodology and analytical framework offer new theoretical references for similar studies in other regions globally.

## Introduction

Global public health governance faces multiple challenges, including the transregional spread of emerging infectious diseases and the increasing burden of chronic diseases. Strengthening the construction of public health systems has become a priority for numerous countries and organizations [1]. In 2023, the WHO emphasized in the Report of the Second Global Technical Consultation on Public Health and Social Measures that regional collaborative development in public health is a critical strategy for addressing cross-border health threats [2]. Collaborative development refers to the process by which multiple stakeholders maximize overall benefits and sustainable development by establishing cooperative mechanisms based on resource advantages and functional roles [3]. At the regional level, collaborative development can dismantle administrative barriers between regions, promoting the free flow and optimal allocation of labor factors across a broader spatial scope. However, narrow national interests [4], global health resource disparities [5], and conflicts in administrative management models [6] negatively affect global public health collaboration to varying degrees. Similarly, regional public health coordination within different countries faces challenges similar to those at the global level but with more localized characteristics.

China's centers for disease control and prevention (CDCs) consist of four levels: national, provincial, municipal, and county. The development of CDCs across

different economic regions remains uneven regarding professional talent structures and equipment resources. Historically, most assistance has taken the form of one-way "blood transfusion"-style support, which fails to establish stable and efficient collaborative development mechanisms [7]. The Chengdu-Chongqing Economic Circle(CCEC), a core area of China's national strategy, encompasses Chongqing Municipality, Chengdu City in Sichuan Province, and their surrounding regions. Given that this region sits within an economically underdeveloped area of the western inland, public health governance capacity within grassroots CDCs is uneven, and talent and technical disadvantages appear even more pronounced. Therefore, by focusing on critical and urgent needs within limited public health resources, grassroots CDCs in the CCEC can establish a collaborative development mechanism tailored to their local context. This approach not only effectively safeguards regional population health but also offers valuable insights for global public health governance.

Current research primarily focuses on macro-level policy design for coordinative development and theoretical elaboration of collaborative models, yet lacks empirical studies from the perspective of grassroots CDCs. Meanwhile, traditional need analysis tools, such as the analytic hierarchy process and weighting method, quantify need priorities solely through single dimensions like "importance level" or "preference degree" [8],overlooking negative effects arising from unmet needs. The Kano model serves as an analytical tool for classifying and prioritizing needs. Its unique advantage lies in capturing positive and negative perceptions of satisfaction, which makes it highly compatible with the dual characteristics of grassroots CDC collaborative development needs: "rigid baseline requirements" and "flexible value-added elements" [9]. This model provides dual decision-making references for public health governance enabling risk avoidance while enhancing value. Therefore, this study investigates grassroots CDC personnel in the CCEC. By applying the Kano model to categorize attributes and quantify priorities of collaborative development needs, this research aims to provide scientific references and a theoretical foundation for public health governance and the establishment of localized collaborative development mechanisms within the CCEC. Simultaneously, it seeks to offer a reference for needs analysis in regional public health collaboration under similar global development contexts.

## Method

### Literature review and Delphi expert consultation method

From December 15, 2024 to December 31, 2024, a literature search was conducted in databases including CNKI, Wanfang, and PubMed. The search employed the keywords "CDC," "high-quality development," "coordinative development," and "integrated development" to retrieve Chinese and English literature. An indicator system for the coordinative development needs of grassroots CDCs was initially drafted, consisting of 6 primary needs and 19 secondary needs (Fig 1).

From January 13, 2025 to February 3, 2025, a Delphi survey was organized. Experts were selected based on the research objectives and principles of representativeness and authority. The selection criteria included: ① at least 5 years of experience in public health, health service management, social medicine, preventive medicine, or related fields; ② a bachelor's degree or above; ③ an associate senior title (associate professor/associate chief physician) or higher. Through random sampling, 13 experts were recruited from the Public Health Management Professional Committee of the Sichuan and Chongqing Preventive Medicine Association. Among these experts, five held associate senior titles (41.7%) and seven held senior titles (58.3%). Their fields of expertise included public health and preventive medicine (n = 9; 75.0%), health service management (n = 2; 16.7%), and social medicine (n = 1; 8.3%). Regarding institutional affiliation, five worked in disease prevention and control centers (41.7%), one in a community health service center (8.3%), four in medical colleges or universities (33.3%), and two in tertiary hospitals (16.7%). Professional experience varied, with two experts having 10–15 years of experience (16.7%), three having 16–20 years (25.0%), and seven having over 20 years (58.3%).

During the first round of the survey, 10 experts (76.9%) provided revision suggestions; consequently, three primary needs were merged, seven secondary needs were deleted, and two secondary needs were added. In the second round, four experts (30.8%) suggested revisions that led to the merging of two secondary needs. The finalized indicator system for the coordinative development needs of grassroots CDCs comprised 4 primary needs and 13 secondary needs. The

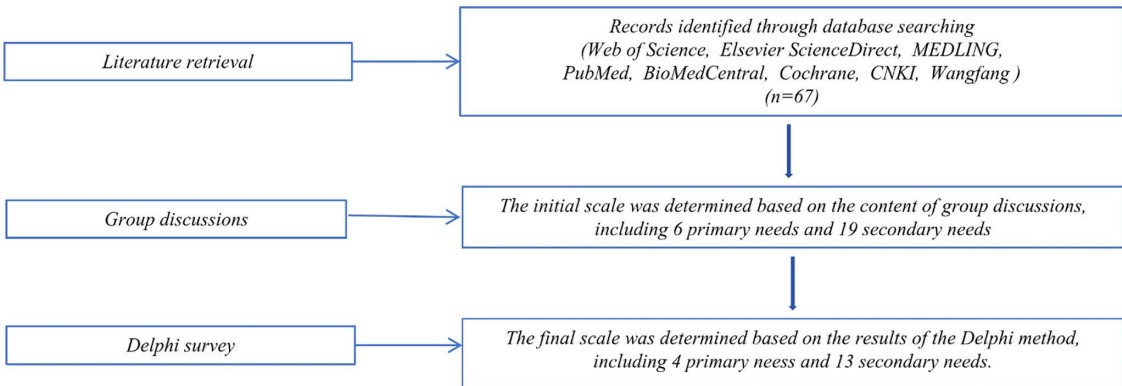

**Fig 1. Flowchart of needs screening.**

expert authority coefficient (Cr) reached 0.90 in the first round and 0.92 in the second, both exceeding the 0.70 threshold for high authority. Kendall's coefficient of concordance for expert opinions was 0.449 and 0.452; both values were statistically significant, indicating a high degree of consensus among experts regarding the survey results (Fig 2, Table 1).

## Questionnaire design

The survey instrument consists of two sections:

(1) Demographic characteristics: This section records participant data, including region, gender, age, administrative level, field of work, and job position.

(2) Kano model questionnaire: This questionnaire sets positive and negative questions for each need. Each question employs a Likert five-point scale, offering five response options: "Like" (1), "Should be" (2), "Does not matter" (3), "Can accept" (4), and "Dislike" (5). By synthesizing the response combinations from the positive and negative questions, each need is categorized according to the Kano two-dimensional attribute classification matrix (Table 2). The attributes include the must-be attribute (M), one-dimensional attribute (O), attractive attribute (A), indifferent attribute (I), and reverse attribute (R). For the must-be attribute, which represents fundamental requirements, satisfaction does not significantly increase when these functions are met, but it drops sharply when they are not. By contrast, satisfaction for the one-dimensional attribute increases proportionally as features are provided and decreases when they are omitted. For the attractive attribute, satisfaction does not decrease if features are absent, but it increases substantially when they are provided. Regarding the indifferent attribute, user satisfaction remains largely unchanged regardless of whether these features are provided. Consequently, the priority order for addressing these needs follows the hierarchy of $M > O > A > I$. Results from the pre-survey demonstrate high reliability and validity for the instrument, with Cronbach's $\alpha = 0.901$, KMO $= 0.943$, and Bartlett's test of sphericity $P < 0.05$.

## Survey methods

The CCEC encompasses 15 cities in Sichuan Province and 27 districts/counties in Chongqing Municipality. Based on economic development and regional location, the CCEC comprises four areas: the Chengdu Metropolitan Area, the Chongqing Metropolitan Area, the Southern Sichuan and Western Chongqing Urban Agglomeration, and the Central Chengdu–Chongqing Urban Agglomeration. To ensure representativeness and scientificity, this study adopted a multi-stage sampling method to identify research subjects. In the first stage (February 13, 2025), a completely random sampling

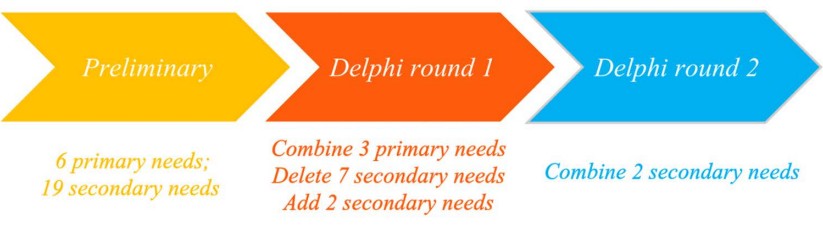

Fig 2. Delphi survey.

**Table 1. Collaborative development needs of grassroots CDCs.**

| Primary needs | Secondary needs | Needs number |
|---|---|---|
| Organizational management and mechanism | Performance-based compensation (1) Permit for medical and health institutions to exceed the current salary control levels for public institutions; (2) Permit for medical service income to be used after deducting costs and extracting various funds, and mainly for personnel rewards) | Q1 |
| | Business archive co-construction and sharing mechanism (emergency response plan and technical solution) | Q2 |
| | Public health emergency rescue mechanism (emergency material reserve and cross-regional material mobilization) | Q3 |
| | Collaborative release and interaction mechanism of social integrated media information | Q4 |
| Handling of public health emergencies | Cross-regional case monitoring, investigation, and tracking | Q5 |
| | Regional monitoring and early-warning information management system | Q6 |
| | Coordinated transfer and diversion diagnosis and treatment of patient with infectious diseases within the region | Q7 |
| | Joint risk assessment and emergency command | Q8 |
| Public health governance | Community chronic disease prevention and services | Q9 |
| | Entity institutions for the collaborative governance of public health | Q10 |
| Talent cultivation | Talent exchange and scientific research innovation cooperation | Q11 |
| | Regional co-construction and sharing between the university and the local area | Q12 |
| | Construction of a cross-regional expert database | Q13 |

**Table 2. Kano model's two-dimensional attribute classification matrix table.**

| Positive questions | Negative questions | | | | |
|---|---|---|---|---|---|
| | Like | Should be | Does not matter | Can accept | Dislike |
| Like | Q | A | A | A | O |
| Should be | R | I | I | I | M |
| Does not matter | R | I | I | I | M |
| Can accept | R | I | I | I | M |
| Dislike | R | R | R | R | Q |

method was used to select one city from each of the four areas, resulting in two sample cities from Sichuan Province and two from Chongqing Municipality. In the second stage (February 17, 2025 to March 3, 2025), the directors of the CDCs in these four cities randomly selected staff members to serve as survey subjects. The inclusion criteria required that participants: ① had worked in a grassroots disease prevention and control center for at least one year; ②participated voluntarily with a commitment to provide honest responses. A total of 117 questionnaires were collected. After excluding responses containing logical inconsistencies or identical options for both positive and negative questions, 110 valid questionnaires retained, yielding an effective rate of 94.02%.

## Ethical consideration

This study obtained the verbal consent of the respondents. Approved by the Ethics Committee of Shapingba District Center for Disease Prevention and Control, the respondents were exempt from signing a written informed consent form. Relevant exemption items have been incorporated into the scope of ethical review, with the approval number 2024121001.

In the introduction of the Delphi expert consultation questionnaire and the Kano model questionnaire, we described in writing to the respondents the research purpose, process and content in detail. At the same time, we reserved sufficient time for the respondents to read the introduction and actively asked if they had any questions, providing detailed and patient answers until the respondents clearly indicated that they fully understood.

Subsequently, one researcher clearly and directly inquired of the respondents whether they agreed to participate in this research, ensuring that their informed consent was given voluntarily, explicitly, and on the basis of full understanding. Meanwhile, another researcher supervised and witnessed the process of verbal consent. After obtaining verbal consent, we immediately provided the respondents with the questionnaire link and explained in detail the operation steps and precautions for filling out the questionnaire to ensure that they could complete it smoothly. If the respondents were emotionally disturbed during the survey, they could withdraw at any time without having to provide a reason.

After the survey, we assured the respondents that all questionnaire data and personal information (such as job title, gender, age) would be strictly confidential and stored in an electronic database that required login account and password (https://www.wjx.cn/). This study did not involve minors under the age of 18.

## Statistical analysis

Statistical analysis was performed using SPSS 25.00. Categorical data appear as frequencies (n) and proportions (%). Based on the Kano model's two-dimensional attribute classification matrix, the better coefficient (SI) and worse coefficient (DSI) for each need were calculated using the method proposed by Berger [10]. The SI represents the satisfaction index following demand fulfillment (Formula 1), whereas the DSI represents the dissatisfaction coefficient resulting from a lack of fulfillment (Formula 2). A four-quadrant scatter plot was generated using SI as the x-axis and the absolute value of DSI as the y-axis for each need attribute. Satisfaction sensitivity assesses the effect of each need on satisfaction within different attributes. The satisfaction sensitivity value (S) is defined as the distance from each need's coordinates to the origin in the quadrant data plot (Formula 3). A higher satisfaction sensitivity indicates that even minimal fulfillment of the need significantly influences overall satisfaction. Finally, the chi-square test was employed to compare need differences among different subgroups.

$$\text{Better coefficient (SI)} = (A + O)/(A + O + M + I);$$

$$\text{Worse coefficient (DSI)} = -1 * (O + M)/(A + O + M + I)$$

$$S = \sqrt{SI^2 + |DSI|^2}$$

# Result

## General situation analysis

Among the 110 respondents, 49 (44.55%) were from Sichuan and 61 (55.45%) were from Chongqing, Regarding gender distribution, 44 cases (40.00%) were male, and 66 cases (60.00%) were females;. Age distribution showed that 42 respondents (38.18%) were aged 30 years or younger, whereas 64 (58.18%) fell within the 31–50 age range. Administratively, 85 participants (77.27%) worked at the district and county level, and 25 (22.73%) worked at or above the municipal level. In terms of professional roles, 90 individuals (81.82%) were engaged in health technology, whereas 20 (18.18%) worked in administrative logistics. Finally, the sample included 77 ordinary employees (70.00%) and 33 department heads or center leaders (30.00%) (Table 3).

## Kano model attribute frequency table and four-quadrant scatter plot

The constituent ratio of the reverse attribute and questionable responses was below 5%. This low frequency indicates that respondents were relatively cautious and consistent when evaluating each potential need (Table 4).

A four-quadrant scatter diagram was constructed using (0.482, 0.586) as the coordinate origin, with the SI value as the y-axis and the absolute value of the DSI as the x-axis. In this distribution, Q2, Q8, and Q12 are located in the first quadrant, representing the one-dimensional attribute. Indicators Q6, Q7, and Q9 fall within the second quadrant, as attractive attributes. Q4, Q10, and Q13 are situated in the third quadrant, identifying them as indifferent attributes. Finally, Q1, Q3, Q5, and Q11 are positioned in the fourth quadrant, representing the must-be attributes (Fig 3)

Table 3. General information of the survey subjects (n, %).

| Demographic characteristics | Number | Percentage (%) |
|---|---|---|
| Region | | |
| Sichuan | 49 | 44.55% |
| Chongqing | 61 | 55.45% |
| Gender | | |
| Male | 44 | 40.00% |
| Female | 66 | 60.00% |
| Age | | |
| ≤30 years old | 42 | 38.18% |
| 31–50 years old | 64 | 58.18% |
| ≥51 years old | 4 | 3.64% |
| Center for disease control and prevention level | | |
| District and county | 85 | 77.27% |
| Municipal level and above | 25 | 22.73% |
| Field of work | | |
| Front-line business | 90 | 81.82% |
| Administrative logistics | 20 | 18.18% |
| Job position | | |
| Department head or center leaders | 33 | 30.00% |
| Ordinary workers | 77 | 70.00% |

**Table 4. Frequency table of collaborative development needs attributes based on the Kano model.**

| Needs number | A | I | M | O | Q | R | SI | DSI |
|---|---|---|---|---|---|---|---|---|
| Q1 | 10.91% | 7.27% | 54.55% | 22.73% | 1.82% | 2.73% | 0.352 | −0.810 |
| Q2 | 9.09% | 13.64% | 34.55% | 38.18% | 3.64% | 0.91% | 0.495 | −0.762 |
| Q3 | 18.18% | 10.91% | 43.64% | 22.73% | 2.73% | 1.82% | 0.429 | −0.695 |
| Q4 | 4.55% | 61.82% | 5.45% | 22.73% | 4.55% | 0.91% | 0.288 | −0.298 |
| Q5 | 16.36% | 13.64% | 47.27% | 18.18% | 3.64% | 0.91% | 0.362 | −0.686 |
| Q6 | 32.73% | 8.18% | 18.18% | 36.36% | 2.73% | 1.82% | 0.724 | −0.571 |
| Q7 | 38.18% | 22.73% | 11.82% | 22.73% | 2.73% | 1.82% | 0.638 | −0.362 |
| Q8 | 10.91% | 13.64% | 27.27% | 43.64% | 1.82% | 2.73% | 0.571 | −0.743 |
| Q9 | 31.82% | 18.18% | 15.45% | 30.00% | 2.73% | 1.82% | 0.648 | −0.476 |
| Q10 | 7.27% | 52.73% | 12.73% | 20.91% | 3.63% | 2.73% | 0.298 | −0.362 |
| Q11 | 16.36% | 13.64% | 38.18% | 27.27% | 2.73% | 1.82% | 0.457 | −0.686 |
| Q12 | 25.45% | 9.09% | 35.45% | 25.45% | 2.73% | 1.82% | 0.533 | −0.638 |
| Q13 | 13.64% | 31.82% | 18.18% | 31.82% | 2.73% | 1.82% | 0.476 | −0.524 |
| Average value | | | | | | | 0.482 | −0.586 |

### Importance ranking of collaborative development needs

Based on the satisfaction sensitivity analysis, the priority ranking of needs follows the hierarchy of M > O > A > I. Within each attribute category, the specific indicators are ranked as follows: for the must-be attribute (M), the order is Q1 > Q11 > Q3 > Q5; for the one-dimensional attribute (O), the order is Q8 > Q2 > Q12; for the attractive attribute (A), the order is Q6 > Q9 > Q7; and for the indifferent attribute (I), the order is Q13 > Q10 > Q4 (Table 5).

### Analysis of need differences among subgroups

Subgroup analysis reveals variations in the Kano attribute classification for specific needs; however, needs that maintain consistent attribute classifications across all subgroups are omitted from Table 6. The results indicate that respondents from Sichuan Province prioritize joint risk assessment and emergency command more significantly ($P = 0.019$). Furthermore, personnel engaged in health technology emphasize the public health emergency rescue mechanism ($P = 0.003$). Within the job position groupings, department heads and center leaders demonstrate a higher priority for mechanisms concerning the coordinated transfer and diversion diagnosis and treatment of patients with infectious diseases within the region, as well as the construction of a cross-regional expert database ($P < 0.05$).

### Discussion

Permitting CDCs to conduct compensated technical services and other activities based on their responsibilities to improve salary and benefits represents the top priority for collaborative needs. This finding reflects the robust appeal among grassroots CDCs within the CCEC for enhanced compensation. Currently, China's CDCs at all levels operate as public welfare institutions and do not provide paid health services. Given that staff salary levels correlate significantly with local fiscal capacity, the eastern part of Sichuan and Chongqing has experienced a 15.3% staff turnover rate [11]. Conversely, CDCs in eastern China benefit from more comprehensive fiscal guarantee systems, allowing them to prioritize digital construction and high-end technological cooperation [12], which creates a significant regional disparity compared to the CCEC. The National Health Service in the UK reduced staff turnover by 18% by establishing a "performance-based salary mechanism" [13]. Consequently, formulating the "Compensated Technical Service Management Measures" under the leadership of the governments of the two regions and in collaboration with the local Finance Bureaus, with reference to the local

**Fig 3. Better-worse scatter plot.**

**Table 5. Importance analysis scale.**

| Needs number | Kano attribute | Satisfaction sensitivity (s) | Satisfaction sensitivity ranking | Priority ranking within attributes |
|---|---|---|---|---|
| Q1 | M | 0.883 | 4 | 1 |
| Q11 | M | 0.824 | 6 | 2 |
| Q3 | M | 0.817 | 7 | 3 |
| Q5 | M | 0.775 | 9 | 4 |
| Q8 | O | 0.937 | 1 | 1 |
| Q2 | O | 0.909 | 3 | 2 |
| Q12 | O | 0.832 | 5 | 3 |
| Q6 | A | 0.922 | 2 | 1 |
| Q9 | A | 0.804 | 8 | 2 |
| Q7 | A | 0.734 | 10 | 3 |
| Q13 | I | 0.708 | 11 | 1 |
| Q10 | I | 0.469 | 12 | 2 |
| Q4 | I | 0.415 | 13 | 3 |

medical service price catalog to establish charging mechanisms for health management and public health monitoring, is of great significance for ensuring the stability of the grassroots CDCs. Furthermore, "talent exchange and scientific research innovation cooperation" constitutes a must-be attribute, reflecting the desire of respondents to enhance their professional

**Table 6. Subgroup analysis.**

| Demographic characteristics | Needs number | Subgroup | Kano attribute | SI | DSI | $\chi^2$ | P |
|---|---|---|---|---|---|---|---|
| Region | Q8 | Sichuan | M | 0.396 | −0.625 | 5.624 | 0.019 |
| | | Chongqing | O | 0.719 | −0.842 | | |
| Field of work | Q3 | Health technology | M | 0.364 | −0.671 | 9.321 | 0.003 |
| | | Administrative logistics | O | 0.765 | −0.823 | | |
| Job position | Q7 | Department head or center leaders | O | 0.767 | −0.433 | 5.978 | 0.016 |
| | | Ordinary workers | A | 0.587 | −0.333 | | |
| | Q13 | Department head or center leaders | O | 0.625 | −0.719 | 4.780 | 0.031 |
| | | Ordinary workers | I | 0.411 | −0.438 | | |

capabilities and scientific innovation capacity. As emerging infectious diseases, chronic illnesses, and environmental hazards escalate globally, the operational landscape of grassroots CDCs has shifted from singular infectious disease prevention to multidimensional governance encompassing infectious diseases, chronic diseases, and environmental health. Traditional knowledge structures centered on preventive medicine struggle to address these complex health challenges [14]. Therefore, introducing interdisciplinary expertise from fields such as computer science and sociology through talent exchange, whereas exploring precision disease prevention and control strategies through technological innovation, holds significant importance. The "Public Health Research Collaboration Network" advocated by the African Research Universities Alliance effectively achieves dual outcomes of talent exchange and scientific research collaboration [15], providing valuable insights for the coordinated development of grassroots CDCs.

The respondents identified five collaborative needs: the public health emergency rescue mechanism, cross-regional case management, joint risk prevention and emergency command, information system construction, and patient triage and treatment. These five aspects collectively form a three-dimensional framework of "basic support–core efficacy–supporting system." The essential requirement for public health rescue mechanisms, encompassing emergency supply reserves and cross-regional resource mobilization, reflects that respondents endured shortages and logistical bottlenecks during the pandemic, and now regard coordinated supply management as a fundamental benchmark for effective collaboration. In 2025, the 194 member states of the WHO jointly signed the "Pandemic Agreement" to ensure equitable access to vaccines and protective equipment based on the principles of access and benefit-sharing [16]. This aligns with the findings of this study, as both reflect attempts to reduce the risk of resource misallocation during public health events through institutionalized coordination of material reserves and distribution. Therefore, the establishment of a collaborative development alliance for CDCs and the implementation of a trans-regional material allocation mechanism—"application by primary CDCs, dual-center review, and green-channel transportation"—with the support of emergency management authorities can effectively enhance the capacity of CDCs in remote areas to respond to public health emergencies. Equally fundamental is the role of "cross-regional case monitoring, investigation, and tracking". Its designation as an essential requirement stems from the inherent tension between territorial management systems and the reality of cross-regional transmission risks. Against the backdrop of globalization and urbanization, the risk of cross-regional spread during infectious disease outbreaks has intensified. China's administrative management system, rooted in "territorial management" at all levels of CDCs, struggles to address the complex challenges of sudden public health emergencies [17]. During the COVID-19 pandemic, 37% of close contacts in Chongqing and Chengdu moved across regions due to delayed information sharing [18]. Addressing this practical pain point, respondents rated "cross-regional case monitoring, investigation, and tracking" as a must-be attribute to establish a "full-chain tracing" defense for regional infectious disease control by standardizing case information and streamlining collaborative investigation processes.

Simultaneously, "joint risk assessment and emergency command" exhibited the highest satisfaction sensitivity (0.937), identifying it as the collaborative development need most capable of significantly enhancing respondent satisfaction. This highlights its core efficacy within the collaborative system. Risk governance theory emphasizes that cognitive consistency is central to the effectiveness of risk prevention and control [19]. The UK's National Public Health Emergency Response Manual defines core indicators for cross-regional risk assessment while establishing a weekly joint conference system for regional CDC directors to ensure sustained cognitive consensus [20]. Therefore, only through unified judgments and joint command regarding risk levels, transmission pathways, and response priorities can grassroots CDCs avoid misaligned prevention measures and disordered resource allocation caused by cognitive biases, thereby enhancing cross-regional coordination efficiency. Consequently, the government-led establishment of the CCEC grassroots disease control collaborative prevention and control joint meeting—implementing monthly regular consultations and temporary consultations for sudden public health events to establish an integrated emergency command platform—has significant practical significance for breaking administrative regional barriers. The "regional monitoring and early warning information management system" and "coordinated transfer and diversion diagnosis and treatment of infectious disease patients within the region" constitutes systematic auxiliary requirements for collaborative development. Once fulfilled, these provide effective support for must-be attributes information backing and clinical coordination. In recent years, as digital governance concepts have risen, big data technology has become an indispensable component in responding to public health emergencies. In China's Yangtze River Delta region, a regional public health information network platform covering 286 disease control institutions enables second-level sharing of case reports and pathogen detection data, facilitating rapid response during the initial stages of emergencies [21], This offers a referenceable direction for the collaborative development of grassroots CDCs in the CCEC. Therefore, formulating the "Specifications for Core Fields of Case Information Sharing" to clarify basic case information, diagnosis and treatment information, the number of close contacts, activity trajectories, and other relevant content is crucial for further optimizing the trans-regional case monitoring and early warning mechanism. Coordinated transfer and diversion diagnosis and treatment of infectious disease patients within the region can be viewed as an extension of the public health emergency response system for cross-regional resource mobilization. By coordinating designated hospital beds and treatment resources across the region, this mechanism prevents medical resource strain caused by patient overcrowding in any single area. These five requirements are not isolated but form an organic whole centered on addressing cross-regional public health risks. This framework is rooted in the practical challenges faced during COVID-19 prevention and control in the CCEC while aligning closely with global public health system reform directions. It provides clear prioritization and a practical pathway for the coordinated development of grassroots CDCs in the CCEC.

Different subgroups of the respondents have varying perceptions of specific collaborative development needs, providing a basis for the precise allocation of public health resources and the implementation of differentiated strategies. Respondents from Sichuan Province demonstrated higher concern for joint risk assessment and emergency command, which likely results from the relatively weak foundation and talent resources of grassroots CDCs in Sichuan [22]; thus, they have a more urgent need for unified cross-regional emergency risk assessment standards and collaborative emergency command. By contrast, Chongqing has initially established a digital application system for infectious disease prevention and control, and the municipal government system under direct jurisdiction has relatively strong capabilities in coordinating health resources [23]. These results suggest that the coordinated development of disease control institutions within the CCEC should consider regional differences, strengthen the standardization of joint risk assessment indicators and the collaborative mechanism of emergency command, and achieve complementary regional advantages.

Respondents engaged in health technology demonstrated greater concern for the public health emergency rescue mechanism. Health technology personnel likely possess a greater awareness of the decisive role that cross-regional material coordination capabilities play in improving emergency handling efficiency compared to those in administrative roles. Therefore, establishing a public health emergency rescue mechanism requires incorporating the opinions of health technology personnel and optimizing the standards for emergency material reserves and cross-regional allocation

processes. Department heads and center leaders demonstrated higher concern for the "mechanism for the diversion and treatment of infectious disease patients within the region" and the "construction of a cross-regional expert database." As decision-makers, these individuals can predict the potential medical resource congestion caused by excessive patient concentration and the constraints on prevention and control efficiency due to technical shortcomings [24]. Ordinary workers, whose daily work mainly involves chronic disease management and public health monitoring, rely more on local talent and have a weaker perception of the need for cross-regional expert support. At the same time, they may fear that relying on external experts might expose their own technical shortcomings, thereby reducing their expectations for this need. However, ignoring the construction of a cross-regional expert database may hide significant prevention and control risks. During the West African monkeypox epidemic [25], the public health institutions in Guinea were unable to conduct virus genotyping due to the lack of cross-regional expert support, leading to misjudgment of the transmission chain and delaying international aid [26]. This indicates that even if ordinary staff have a low awareness of this demand, building a multidisciplinary cross-regional expert database remains a vital forward-looking measure to enhance the resilience of public health emergency response [27].

Different practice paradigms have emerged in global public health governance, with the comparative effectiveness of "formal collaborative entities" versus "informal collaborative networks" remaining a central point of contention. The "European Center for Disease Prevention and Control" has achieved standardized sharing of epidemic data and unified resource scheduling among member states by establishing permanent regional collaborative entities, which offers the advantage of clear responsibility [28]. Conversely, the informal network based on "informational joint meetings" used by the "Regional Health Emergency Alliance" in the US responds more flexibly to the needs of grassroots regions [29]. This study identifies "public health collaborative governance entities" as an indifferent attribute, which does not imply the need is unimportant but rather reflects the rational choice of grassroots CDCs in the CCEC based on informal mechanisms under the Chinese political structure. Because CCEC grassroots CDCs work under different local governments, establishing a formal collaborative entity might change direct connections into multilevel government reporting, which could lead to distrust of top-down reform and reduce efficiency. This is consistent with conclusions proposed in Southeast Asian public health governance research [30]. Moreover, "social integrated media information collaborative release and interaction" is regarded as an indifferent attribute. This may result from the law of diminishing marginal utility [31], where key information becomes diluted when overly embedded in social interaction scenarios. The weak satisfaction perception for this demand reflects that public health information collaboration should not prioritize the diversification of forms but should instead precisely match core health needs. In the digital era, social integrated media has become the primary carrier for popularizing health information, and its communication effectiveness can directly affect the health literacy level of the population. Therefore, achieving the core objectives of health education through precise content delivery and appropriate interactive engagement represents a vital developmental direction for social converged media in public health governance.

## Limitation

This study has several limitations. First, the respondents were solely from the Sichuan and Chongqing regions, which may lead to selection bias. Therefore, these research results only serve as a preliminary reference for the formulation of collaborative public health policies within the CCEC and cannot be directly used as a decision-making basis for governments at all levels. Future research should include cross-regional comparative studies with areas such as the Yangtze and Pearl River Deltas to analyze commonalities and differences in grassroots CDCs needs across different economic levels, thereby providing a more comprehensive reference for the construction of the national public health system. Second, this study utilized only the Kano model to conduct a quantitative assessment of potential needs. Although this method clarifies attribute classifications, it fails to reveal the underlying mechanisms that form these needs. Subsequent research should incorporate qualitative methods, such as semistructured interviews and field investigations, to explore the deep-seated factors influencing the emergence of different needs.

## Conclusion

This study marks the first application of the Kano model to assess regional collaborative development needs in CDCs. Using the better-worse coefficient method, 13 potential needs were categorized into 4 must-be attributes, 3 one-dimensional attributes, 3 attractive attributes, and 3 indifferent attributes. Combined with satisfaction sensitivity analysis, this approach provides a quantifiable tool for prioritizing public health needs. Although the findings exhibit regional limitations, the research methodology and analytical framework offer novel theoretical references for similar studies in other domestic and international regions.

## Supporting information

**S1 File. Questionnaire and data analysis process.**
(ZIP)

## Acknowledgments

The authors thank the experts and staff members who participated in the questionnaires and interviews.

## Author contributions

**Conceptualization:** Yang Tong, Zhang Min.

**Data curation:** Yang Tong, Tan Bo, Zhang Min.

**Formal analysis:** Yang Tong, Huang Qianzhen, Hu Bin.

**Funding acquisition:** Yang Tong, Huang Qianzhen, Tan Bo, Hu Bin.

**Investigation:** Tan Bo.

**Methodology:** Huang Qianzhen, Tan Bo, Hu Bin.

**Project administration:** Hu Bin.

**Software:** Hu Bin.

**Validation:** Yang Tong.

**Visualization:** Yang Tong, Zhang Min.

**Writing – original draft:** Yang Tong, Huang Qianzhen, Zhang Min.

**Writing – review & editing:** Zhang Min.

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
