## [Decision Letter · Decision Letter 0]

29 Jan 2026

PLOS One

Dear Dr. Min,

Thank you for submitting your manuscript to PLOS ONE. After careful consideration, we feel that it has merit but does not fully meet PLOS ONE’s publication criteria as it currently stands. Therefore, we invite you to submit a revised version of the manuscript that addresses the points raised during the review process.

We look forward to receiving your revised manuscript.

Kind regards,

Wuqi Qiu

Academic Editor

PLOS One

**Journal Requirements:**

2. In the ethics statement in the Methods, you have specified that verbal consent was obtained. Please provide additional details regarding how this consent was documented and witnessed, and state whether this was approved by the IRB.

“This study was financially supported by the Chongqing Science and Health Joint Medical Research Project (2024MSXM172), and Shapingba District Science and Health Joint Medical Research Project (2024SQKWLHMS032), and Shapingba District Science and Health Joint Medical Research Project (2025SQKWLHMS072).”

4. Please note that funding information should not appear in any section or other areas of your manuscript. We will only publish funding information present in the Funding Statement section of the online submission form. Please remove any funding-related text from the manuscript.

5. In the online submission form, you indicated that “The data that support the findings of this study are available from the authors but restrictions apply to the availability of these data, which were used under license from the experts involved in the Delphi survey and the staff of CDCs who participated in the questionnaire survey for the current study, and so are not publicly available. However, data are available from the authors upon reasonable request and with permission from these experts and respondents.”

7. Please remove your figures from within your manuscript file, leaving only the individual TIFF/EPS image files, uploaded separately. These will be automatically included in the reviewers’ PDF.

8. If tables are embedded in the manuscript and ALSO loaded as separate files, please delete the separate files, leaving only the tables in the manuscript file.

9. We note that the data set contains text or data that is not in English. Please note that PLOS is an English-language publisher, so we require data sets to be provided in English as well. Please upload an English-language version of your data set.

This will also allow us to determine if your data follows PLOS standards per our Data Availability policy here: https://journals.plos.org/plosone/s/data-availability

**Additional Editor Comments:**

We do agree that the topic is important and the Journal audience deserves discussion of this topic, so please consider revision and resubmission.  Please revise this manuscript according to the comments of two reviewers.

Reviewers' comments:

Reviewer's Responses to Questions

**Comments to the Author**

1. Is the manuscript technically sound, and do the data support the conclusions?

Reviewer #1: Partly

Reviewer #2: Yes

2. Has the statistical analysis been performed appropriately and rigorously?

Reviewer #1: Yes

Reviewer #2: Yes

3. Have the authors made all data underlying the findings in their manuscript fully available?

Reviewer #1: Yes

Reviewer #2: Yes

4. Is the manuscript presented in an intelligible fashion and written in standard English?

Reviewer #1: Yes

Reviewer #2: Yes

Reviewer #1: This paper focuses on the coordinated development of disease prevention and control regions, and it holds great research significance.

1.The description of the Delphi research method is too simplistic. It is suggested to include the process of expert selection and indicator construction.

2. Kano sampled four cities. Please provide a representative description of the sampling cities. The description mentioned as "multistage sample" merely indicates the sequence of sampling surveys conducted by the institution leaders and other personnel. The analysis does not reflect the differences in results among different subgroups, resulting in insufficient depth of the results.

3. "Two permits" is a policy term. It is recommended to provide a definition of the term at the first mention and not to explain it until the discussion section.

4. In terms of English expression: Simplify sentences, unify terminology, and complete the references in accordance with the journal requirements.

Reviewer #2: I. Overall Evaluation

This study focuses on the collaborative development needs of grassroots Centers for Disease Control and Prevention (CDCs) in the Chengdu-Chongqing Economic Circle, innovatively applying the Kano Model to the analysis of regional collaborative needs in the public health field. The research topic aligns with global public health governance priorities and holds significant practical value and theoretical reference significance. Overall, the paper has a complete structure and clear logic with a relatively standardized research design. However, there is room for improvement in format details, language expression, methodological refinement, and discussion expansion. Detailed comments on specific dimensions are as follows:

II. Specific Review Dimensions

(I) Compliance with Paper Format Requirements

Conclusion: Basically compliant with PLOS ONE's submission format requirements, but with some non-standard details.

Detailed Explanation:

1. Strengths: The paper includes complete modules such as title, abstract, introduction, methods, results, discussion, conclusion, acknowledgments, author contributions, conflict of interest statement, ethics and consent, funding information, data availability, ORCID, and references, conforming to the standard structure of academic papers. Tables and figures are numbered standardizedly (e.g., Figure 1, Table 1) and accurately cited in the main text, with unified data presentation formats (e.g., categorical data expressed as frequency and proportion).

2. Weaknesses:

- Layout issues in tables: The content layout of "Table 5. Importance analysis scale" is chaotic, with unclear correspondence between column headers and data, affecting readability.

- Insufficient clarity in the hierarchy of some section headings: For example, the subheadings in the "Methods" section do not strictly follow the "Level 1 Heading - Level 2 Heading" hierarchical logic (e.g., "Literature Review and Delphi Expert Consultation Method" and "Questionnaire Design" should be peer subheadings with inconsistent formats).

(II) Adequacy of Academic English for Publication

Conclusion: The English language basically meets the needs of academic communication but has issues such as grammatical errors, redundant expressions, and inconsistent terminology use, failing to fully meet PLOS ONE's academic English publication standards.

Detailed Explanation:

1. Strengths: Core academic terms (e.g., "Kano Model", "must-be attribute", "Better-Worse coefficient") are used accurately, and sentence structures basically conform to academic English expression habits, enabling clear communication of research ideas and results.

2. Weaknesses:

- Grammatical and syntactic issues: Some sentences have subject-verb disagreement, tense confusion, and other errors (e.g., in "the capacity for public health governance varies significantly among CDCs across different countries and regions, with grassroots CDCs facing particular disadvantages", "facing" should be changed to "face" to maintain syntactic consistency). Some long sentences are logically redundant; for example, "Establishing stable and efficient collaborative development mechanisms among CDCs across diverse regions to maximize overall effectiveness and ensure sustainable development has emerged as a critical public health science issue" can be simplified for readability improvement.

- Inconsistent terminology use: For example, "social financial media information" appears multiple times in the text with an inaccurate expression; it should be "social integrated media information" (consistent with the secondary need expression in the table), resulting in contradictions in terminology.

- Non-idiomatic expressions: Some phrases do not conform to common academic English usage. For example, "provide scientific reference and theoretical basis" can be optimized to "provide scientific references and a theoretical foundation", and "strengthen public health systems" can be adjusted to "strengthen the construction of public health systems" to better fit the academic English context.

(III) Appropriateness of Methodology and Rationale

Conclusion: The methodology selection is appropriate and the design is reasonable, aligning with the research objectives. However, some details lack sufficient explanation and can be further refined.

Detailed Explanation:

1. Appropriateness Analysis:

- High alignment between research methods and objectives: To address the core goal of "classifying and prioritizing collaborative development needs of grassroots CDCs", a combined method of "literature review + Delphi expert consultation + Kano Model" is adopted. The literature review and Delphi method ensure the scientific of the demand indicator system (expert authority coefficient Cr > 0.7, significant Kendall's coefficient of concordance), while the Kano Model's two-dimensional attribute classification advantage compensates for the defect of traditional demand analysis tools (e.g., AHP) that ignore the "negative impacts of unmet needs", enabling comprehensive capture of the different attribute characteristics of needs. Moreover, Delphi method was not clearly and explicitely descripted in the methodolgy section.

- Reasonable sampling method and sample size: A multi-stage sampling method was used to select 8 grassroots CDCs from 4 cities in the Chengdu-Chongqing Economic Circle, ultimately obtaining 110 valid questionnaires (effective recovery rate of 94.02%). The sample size meets the basic requirements of Kano Model analysis, and the sample covers respondents from different regions, ages, and positions, with certain representativeness.

- Standard statistical analysis methods: SPSS was used for data processing, and the Better-Worse coefficient method was employed to calculate SI, DSI, and satisfaction sensitivity. The formula derivation is clear, and the selection of statistical test methods is appropriate (e.g., Bartlett's sphericity test for reliability and validity analysis).

2. Suggestions for Improvement:

- Insufficient detailed explanation of the Delphi expert consultation method: The specific selection criteria for experts (e.g., professional title, research field, years of experience) are not clearly defined; only "13 experts were selected from the Public Health Management Special Committee of the Sichuan and Chongqing Preventive Medicine Associations" is mentioned, lacking detailed descriptions of expert backgrounds, which affects the credibility of the methodology.

- The design logic of the Kano Model questionnaire can be further supplemented: The specific assignment rules for each option (e.g., "like it very much", "should be") in the "five-point Likert scale" and the operational process of attribute classification based on positive and negative answers are not explained, which is not conducive to readers replicating and verifying the research results.

- Insufficient demonstration of sample representativeness: It is not analyzed whether the distribution of the sample in demographic characteristics such as "work field" and "education level" is consistent with the overall situation of grassroots CDCs in the Chengdu-Chongqing Economic Circle, nor is the existence of sampling bias and corresponding countermeasures explained.

(IV) Clarity of Results Presentation and Alignment with Research Theme

Conclusion: The research results are basically presented clearly, with core data highly relevant to the research theme. However, the interpretation of some results is not in-depth enough, and the auxiliary explanatory effect of tables and figures needs to be improved.

Detailed Explanation:

1. Strengths:

- Clear classification of results: The results are presented in the logic of "general situation analysis → Kano Model attribute classification → priority ranking of needs", progressing layer by layer and closely focusing on the core theme of "analysis of collaborative development needs".

- Highlighting core data: The priority ranking of needs and satisfaction sensitivity values for different attributes (must-be, one-dimensional, attractive, indifferent) are clearly presented. For example, "performance compensation (0.883)" has the highest priority among must-be attributes, and "joint risk assessment and emergency command (0.937)" has the highest satisfaction sensitivity among one-dimensional attributes, providing solid data support for the analysis in the discussion section.

- The four-quadrant scatter plot (Figure 2) intuitively shows the attribute classification of each need, helping readers quickly understand the characteristics of different needs.

2. Weaknesses:

- Overly brief interpretation of results: Only data rankings are listed, without preliminary explanation of the internal logic of some key results. For example, "talent exchange and scientific research innovation cooperation" appears in both must-be attributes and one-dimensional attributes, but the reasons for this phenomenon (e.g., differences in perceptions of the need among different respondent groups) are not explained.

- Unclear presentation of table data: In "Table 4. Frequency table of collaborative development needs attributes based on the Kano Model", the column alignment of some data (e.g., Q3, Q5) is chaotic, and the proportional values of attributes such as A, I, and M are not clearly corresponding.

- Lack of statistical significance analysis of results: For example, whether there are differences in the perception of need attributes among respondents from different regions (Sichuan vs. Chongqing) and different positions (ordinary staff vs. leaders) has not been analyzed through inter-group comparison, resulting in insufficient richness of results.

(V) Sufficiency of Discussion and Practical Guidance Significance

Conclusion: The discussion section focuses on core results with a certain depth, but there are problems such as a narrow discussion scope, insufficiently specific practical guidance significance, and inadequate dialogue with existing research.

Detailed Explanation:

1. Strengths:

- Closely linking to research results: For the core needs of must-be, one-dimensional, attractive, and indifferent attributes, analysis is conducted by combining relevant research at home and abroad (e.g., the performance-based pay system of the UK's NHS, the collaboration network of the African Research Universities Alliance), explaining the realistic background of demand formation (e.g., the prominent demand for performance compensation due to the high turnover rate of grassroots CDC personnel).

- A three-dimensional collaborative demand framework of "basic guarantee - core operational capacity - supporting system" is refined, providing a theoretical model for the collaborative development of regional CDCs.

2. Weaknesses:

- Insufficient dialogue with existing research: Only some relevant literature is cited for corroboration, without systematically sorting out the research status in the field, nor clearly defining the differences and innovations between this study and existing research (e.g., methodological differences from studies on the collaborative development of grassroots CDCs in other regions, reasons for differences in results).

- Insufficiently specific practical guidance significance: The proposed suggestions are relatively macro (e.g., "establish a coordinated supplementary funding mechanism", "build a regional monitoring and early warning information management system"), without providing operable implementation paths combined with the localized characteristics of the Chengdu-Chongqing Economic Circle such as the administrative system and resource endowments (e.g., the source of funds for the funding mechanism, the construction subject of the information system, and data sharing standards).

- Insufficiently in-depth discussion of indifferent attributes: Only the surface reasons why needs such as "construction of a cross-regional expert database" are classified as indifferent attributes are explained, without further analyzing how to transform such "potential needs" into "effective needs" through policy guidance and mechanism innovation, which limits the practical value of the research.

- Failure to respond to the impact of research limitations on results: For example, the limitation of sample size prevents in-depth subgroup analysis, and it is not discussed whether this limitation will affect the reliability of demand ranking, nor are specific directions for future research proposed.

III. Summary and Revision Suggestions

(I) Overall Summary

This study has clear research value and innovations, with a reasonable methodological design and reliable results, basically meeting PLOS ONE's submission standards. However, comprehensive revisions are needed in format standardization, language polishing, supplementation of methodological details, deepening of result interpretation, and expansion of discussions.

(II) Specific Revision Suggestions

1. Format Optimization:

- Unify reference formats, supplement complete literature information (e.g., page numbers, DOIs) in strict accordance with PLOS ONE's requirements, and ensure consistent formatting.

- Reformat chaotic tables (e.g., Table 5) to clarify the correspondence between column headers and data, improving table readability.

- Standardize the hierarchy of section headings and unify the format of subheadings (e.g., font, indentation, numbering).

2. Language Revision:

- Engage professional academic English editors to polish the full text, correct grammatical errors, optimize sentence structures, and ensure concise, accurate, and idiomatic expression.

- Unify terminology use (e.g., uniformly change "social financial media information" to "social integrated media information") to avoid contradictions.

3. Methodological Supplementation:

- Clearly explain the specific selection criteria (e.g., professional title, years of experience, research field) and background information of Delphi experts to enhance methodological credibility.

- Supplement the assignment rules of the Kano Model questionnaire and the specific operational process of attribute classification to facilitate readers' replication and verification.

- Strengthen the demonstration of sample representativeness, analyze the consistency between sample distribution and the overall situation, and explain countermeasures for sampling bias.

4. Optimization of Results Presentation:

- Supplement preliminary interpretations of key results (e.g., the reasons for "talent exchange and scientific research innovation cooperation" appearing across attributes).

- Correct table layout errors to ensure clear and readable data.

- Add inter-group comparison analysis (e.g., differences in needs among respondents from different regions and positions) to enrich the dimensions of results.

5. Discussion Expansion:

- Systematically sort out relevant research at home and abroad, clarify the innovations and limitations of this study, and strengthen dialogue with existing research.

- Combine the localized characteristics of the Chengdu-Chongqing Economic Circle to propose specific and operable practical suggestions (e.g., specific paths for the funding mechanism and information system construction).

- Deepen the discussion on indifferent attributes and analyze the possibility and paths of transforming them into effective needs.

.

Reviewer #1: No

Reviewer #2: No

---

## [Author Response · Author response to Decision Letter 1]

17 Mar 2026

Response to Editor and Reviewers

Dear Editor and Reviewers:

Thank you for taking time out of your busy schedule to review the manuscript.

When I saw so many practical opinions, I was deeply moved. Currently, If there are any revisions that are not perfect, we will definitely improve them again as soon as possible.

The revisioninstructions are as follows:

To Editor

1.Editor: In the ethics statement in the Methods, you have specified that verbal consent was obtained. Please provide additional details regarding how this consent was documented and witnessed, and state whether this was approved by the IRB.

Reply: Additional details regarding verbal consent are described in detail in the Ethical Considerations section of the manuscript, and it is noted that this study was approved by the Ethics Committee of Shapingba District Center for Disease Control and Prevention.

2.Reviewer: Thank you for stating the following financial disclosure:

“This study was financially supported by the Chongqing Science and Health Joint Medical Research Project (2024MSXM172), and Shapingba District Science and Health Joint Medical Research Project (2024SQKWLHMS032), and Shapingba District Science and Health Joint Medical Research Project (2025SQKWLHMS072).”

Reply: The roles played by the three projects in this study are described in detail in the cover letter. Specifically, the three grants provided financial support for this study, enabling us to cover the labor costs of experts participating in the Delphi survey and the fees associated with manuscript publication. We have included the project information in the cover letter.

3.Reviewer: Please note that funding information should not appear in any section or other areas of your manuscript. We will only publish funding information present in the Funding Statement section of the online submission form. Please remove any funding-related text from the manuscript.

Reply: Thank you for your reminder. We have removed the funding information from the manuscript and added it to the cover letter.

4.Editor: In the online submission form, you indicated that “The data that support the findings of this study are available from the authors but restrictions apply to the availability of these data, which were used under license from the experts involved in the Delphi survey and the staff of CDCs who participated in the questionnaire survey for the current study, and so are not publicly available. However, data are available from the authors upon reasonable request and with permission from these experts and respondents.”

Reply: We have submitted the dataset during the revision process.

5.Editor: Your ethics statement should only appear in the Methods section of your manuscript. If your ethics statement is written in any section besides the Methods, please move it to the Methods section and delete it from any other section. Please ensure that your ethics statement is included in your manuscript, as the ethics statement entered into the online submission form will not be published alongside your manuscript.

Reply: Thank you for your reminder. The ethical statement appears solely in the "Methods" section of the manuscript.

6.Editor: Please remove your figures from within your manuscript file, leaving only the individual TIFF/EPS image files, uploaded separately. These will be automatically included in the reviewers’ PDF.

Reply: We have removed the figures from the manuscript and uploaded separate TIFF/EPS image files.

7.Editor: If tables are embedded in the manuscript and ALSO loaded as separate files, please delete the separate files, leaving only the tables in the manuscript file.

Reply: We have removed the separate table files and retained only the tables in the manuscript.

8.Editor: We note that the data set contains text or data that is not in English. Please note that PLOS is an English-language publisher, so we require data sets to be provided in English as well. Please upload an English-language version of your data set.

This will also allow us to determine if your data follows PLOS standards per our Data Availability policy here: https://journals.plos.org/plosone/s/data-availability.

Reply: All of our datasets have been uploaded in English format.

9.Editor: If the reviewer comments include a recommendation to cite specific previously published works, please review and evaluate these publications to determine whether they are relevant and should be cited. There is no requirement to cite these works unless the editor has indicated otherwise.

Reply: The reviewers did not suggest specific reference documents. However, in response to the reviewers' comments, we have further revised the Discussion section and attached the reference list.

To Reviewer 1

1.Reviewer: The description of the Delphi research method is too simplistic. It is suggested to include the process of expert selection and indicator construction.

Reply: Thanks for your thoughtful comment. We have provided a detailed re-description of the process of literature search and the Delphi survey. In the literature search, we re-reported the sources of the needs as well as their details. In the Delphi process, we added the sources and detailed backgrounds of the experts. At the same time, we also re-described the changes of the original need system in the two rounds of the survey, and re-drew the flowchart to facilitate a better understanding.

Figure 1. Flowchart of needs screening

Figure 2. Delphi survey

From December 15, 2024 to December 31, 2024, a literature search was conducted in databases including CNKI, Wanfang, and PubMed. The search employed the keywords “CDC,” “high-quality development,” “coordinative development,” and “integrated development” to retrieve Chinese and English literature. An indicator system for the coordinative development needs of grassroots CDCs was initially drafted, consisting of 6 primary needs and 19 secondary needs (Figure 1).

From January 13, 2025 to February 3, 2025, a Delphi survey was organized. Experts were selected based on the research objectives and principles of representativeness and authority. The selection criteria included: ① at least 5 years of experience in public health, health service management, social medicine, preventive medicine, or related fields; ② a bachelor’s degree or above; ③ an associate senior title (associate professor/associate chief physician) or higher. Through random sampling, 13 experts were recruited from the Public Health Management Professional Committee of the Sichuan and Chongqing Preventive Medicine Association. Among these experts, five held associate senior titles (41.67%) and seven held senior titles (58.33%). Their fields of expertise included public health and preventive medicine (n = 9; 75.00%), health service management (n = 2; 16.67%), and social medicine (n = 1; 8.33%). Regarding institutional affiliation, five worked in disease prevention and control centers (41.67%), one in a community health service center (8.33%), four in medical colleges or universities (33.33%), and two in tertiary hospitals (16.67%). Professional experience varied, with two experts having 10–15 years of experience (16.67%), three having 16–20 years (25.00%), and seven having over 20 years (58.33%).

During the first round of the survey, 10 experts (76.92%) provided revision suggestions; consequently, three primary needs were merged, seven secondary needs were deleted, and two secondary needs were added. In the second round, four experts (30.77%) suggested revisions that led to the merging of two secondary needs. The finalized indicator system for the coordinative development needs of grassroots CDCs comprised 4 primary needs and 13 secondary needs. The expert authority coefficient (Cr) reached 0.90 in the first round and 0.92 in the second, both exceeding the 0.70 threshold for high authority. Kendall’s coefficient of concordance for expert opinions was 0.449 and 0.452; both values were statistically significant, indicating a high degree of consensus among experts regarding the survey results (Figure 2, Table 1).

2.Reviewer: Kano sampled four cities. Please provide a representative description of the sampling cities.

Reply: Following your suggestion, we have provided more detailed descriptions of the sample cities.

The CCEC encompasses 15 cities in Sichuan Province and 27 districts/counties in Chongqing Municipality. Based on economic development and regional location, the CCEC comprises four areas: the Chengdu Metropolitan Area, the Chongqing Metropolitan Area, the Southern Sichuan and Western Chongqing Urban Agglomeration, and the Central Chengdu–Chongqing Urban Agglomeration. To ensure representativeness and scientificity, this study adopted a multistage sampling method to identify research subjects. In the first stage (February 13, 2025), a completely random sampling method was used to select one city from each of the four areas, resulting in two sample cities from Sichuan Province and two from Chongqing Municipality. In the second stage (February 17, 2025 to March 3, 2025), the directors of the CDCs in these four cities randomly selected staff members to serve as survey subjects.

3.Reviewer: The description mentioned as "multistage sample" merely indicates the sequence of sampling surveys conducted by the institution leaders and other personnel.

Reply: Thank you very much for your candid advice. Due to the unclear description, your misunderstanding occurred. The multi-stage sampling method used in this study was carried out in two stages. The first stage involved randomly selecting cities, and the second stage involved the selection of respondents by the institution leaders. We have re-described the multi-stage sampling process to make it easier to understand.

4.Reviewer: The analysis does not reflect the differences in results among different subgroups, resulting in insufficient depth of the results.

Reply: Thank you very much for your forward-looking suggestions. Based on your advice, we conducted a subgroup analysis of the data and further described the possible causes of the differences in the discussion section of the manuscript.

Result：Subgroup analysis reveals variations in the Kano attribute classification for specific needs; however, needs that maintain consistent attribute classifications across all subgroups are omitted from Table 6. The results indicate that respondents from Sichuan Province prioritize joint risk assessment and emergency command more significantly (P = 0.019). Furthermore, personnel engaged in health technology emphasize the public health emergency rescue mechanism (P = 0.003). Within the job position groupings, department heads and center leaders demonstrate a higher priority for mechanisms concerning the coordinated transfer and diversion diagnosis and treatment of patients with infectious diseases within the region, as well as the construction of a cross-regional expert database (P < 0.05).

Table 6. Subgroup analysis

Demographic characteristics Needs

number Subgroup Kano

attribute SI DSI P

Region Q8 Sichuan M 0.396 −0.625 5.624 0.019

Chongqing 0 0.719 −0.842

Field of work Q3 Health technology M 0.364 −0.671 9.321 0.003

Administrative logistics O 0.765 −0.823

Job position Q7 Department head or center leaders O 0.767 −0.433 5.978 0.016

Ordinary workers A 0.587 −0.333

Q13 Department head or center leaders O 0.625 −0.719 4.780 0.031

Ordinary workers I 0.411 −0.438

Discussion:Different subgroups of the respondents have varying perceptions of specific collaborative development needs, providing a basis for the precise allocation of public health resources and the implementation of differentiated strategies. Respondents from Sichuan Province demonstrated higher concern for joint risk assessment and emergency command, which likely results from the relatively weak foundation and talent resources of grassroots CDCs in Sichuan [22]; thus, they have a more urgent need for unified cross-regional emergency risk assessment standards and collaborative emergency command. By contrast, Chongqing has initially established a digital application system for infectious disease prevention and control, and the municipal government system under direct jurisdiction has relatively strong capabilities in coordinating health resources [23]. These results suggest that the coordinated development of disease control institutions within the CCEC should consider regional differences, strengthen the standardization of joint risk assessment indicators and the collaborative mechanism of emergency command, and achieve complementary regional advantages.

Respondents engaged in health technology demonstrated greater concern for the public health emergency rescue mechanism. Health technology personnel likely possess a greater awareness of the decisive role that cross-regional material coordination capabilities play in improving emergency handling efficiency compared to those in administrative roles. Therefore, establishing a public health emergency rescue mechanism requires incorporating the opinions of health technology personnel and optimizing the standards for emergency material reserves and cross-regional allocation processes. Department heads and center leaders demonstrated higher concern for the “mechanism for the diversion and treatment of infectious disease patients within the region” and the “construction of a cross-regional expert database.” As decision-makers, these individuals can predict the potential medical resource congestion caused by excessive patient concentration and the constraints on prevention and control efficiency due to technical shortcomings [24]. Ordinary workers, whose daily work mainly involves chronic disease management and public health monitoring, rely more on local talent and have a weaker perception of the need for cross-regional expert support. At the same time, they may fear that relying on external experts might expose their own technical shortcomings, thereby reducing their expectations for this need. However, ignoring the construction of a cross-regional expert database may hide significant prevention and control risks. During the West African monkeypox epidemic[25], the public health institutions in Guinea were unable to conduct virus genotyping due to the lack of cross-regional expert support, leading to misjudgment of the transmission chain and delaying international aid [26]. This indicates that even if ordinary staff have a low awareness of this demand, building a multidisciplinary cross-regional expert database remains a vital forward-looking measure to enhance the resilience of public health emergency response [27].

5.Reviewer: "Two permits" is a policy term. It is recommended to provide a definition of the term at the first mention and not to explain it until the discussion section.

Reply: Thanks for your thoughtful comment. When the "two permits" policy was first introduced, an explanation was given for this policy-related term.

Performance-based compensation. (1) Permit for medical and health institutions to exceed the current salary control levels for public institutions; (2) Permit for medical service income to be used after deducting costs and extract

---

## [Decision Letter · Decision Letter 1]

5 Apr 2026

Analyzing the collaborative development needs of grassroots centers for disease control and prevention using the Kano model: A case study of China’s Chengdu–Chongqing Economic Circle

PONE-D-25-68754R1

Dear Ms. Zhang Min,

We’re pleased to inform you that your manuscript has been judged scientifically suitable for publication and will be formally accepted for publication once it meets all outstanding technical requirements.

Kind regards,

Wuqi Qiu

Academic Editor

PLOS One

Additional Editor Comments (optional):

Reviewers' comments:

Reviewer's Responses to Questions

**Comments to the Author**

Reviewer #1: All comments have been addressed

Reviewer #2: All comments have been addressed

2. Is the manuscript technically sound, and do the data support the conclusions?

Reviewer #1: Yes

Reviewer #2: Yes

3. Has the statistical analysis been performed appropriately and rigorously?

Reviewer #1: Yes

Reviewer #2: Yes

4. Have the authors made all data underlying the findings in their manuscript fully available?

Reviewer #1: Yes

Reviewer #2: Yes

5. Is the manuscript presented in an intelligible fashion and written in standard English?

Reviewer #1: Yes

Reviewer #2: Yes

Reviewer #1: In subsequent related research, expand the research area, enhance the representativeness of the survey questionnaire; improve the English expression level.

Reviewer #2: The authors have carefully revised the manuscript following all reviewer comments. The key limitations and weaknesses pointed out in the initial review have been fully eliminated or satisfactorily fixed. On this basis, I confirm that the revised manuscript satisfies the standards for publication and recommend acceptance

.

Reviewer #1: **Yes:** ZhangyingZhangyingZhangyingZhangying

Reviewer #2: No

---

## [Editor Report · Acceptance letter]

PONE-D-25-68754R1

PLOS One

Dear Dr. Min,

I'm pleased to inform you that your manuscript has been deemed suitable for publication in PLOS One. Congratulations! Your manuscript is now being handed over to our production team.

Kind regards,

on behalf of

Dr. Wuqi Qiu

Academic Editor

PLOS One